# A Reinforcement Learning-Based Approach to Automate the Electrochromic Glass and to Enhance the Visual Comfort

**Raghuram Kalyanam \* and Sabine Hoffmann \***

Faculty of Civil Engineering, University of Kaiserslautern, Paul-Ehrlich-Str. 14, 67663 Kaiserslautern, Germany
\* Correspondence: raghuramkalyanam@gmail.com (R.K.); Sabine.Hoffmann@bauing.uni-kl.de (S.H.)

**Abstract:** Daylight is important for the well-being of humans. Therefore, many office buildings use large windows and glass facades to let more daylight into office spaces. However, this increases the chance of glare in office spaces, which results in visual discomfort. Shading systems in buildings can prevent glare but are not effectively adapted to changing sky conditions and sun position, thus losing valuable daylight. Moreover, many shading systems are also aesthetically unappealing. Electrochromic (EC) glass in this regard might be a better alternative, due to its light transmission properties that can be altered when a voltage is applied. EC glass facilitates zoning and also supports control of each zone separately. This allows the right amount of daylight at any time of the day. However, an effective control strategy is still required to efficiently control EC glass. Reinforcement learning (RL) is a promising control strategy that can learn from rewards and penalties and use this feedback to adapt to user inputs. We trained a Deep Q learning (DQN) agent on a set of weather data and visual comfort data, where the agent tries to adapt to the occupant's feedback while observing the sun position and radiation at given intervals. The trained DQN agent can avoid bright daylight and glare scenarios in 97% of the cases and increases the amount of useful daylight up to 90%, thus significantly reducing the need for artificial lighting.

**Keywords:** daylighting; DQN; reinforcement learning; glare; electrochromic glass; radiance; visual comfort; simulations

## 1. Introduction

Climate change and global warming are no longer foreign words in the world today. Their affects are more complex than just climbing temperatures. The latest report from the climate change performance index (CCPI) [1] states that, still, no country is compatible enough with the agreed Paris agreement goals. The latest UN global status report [2] states that buildings contribute up to 39% greenhouse gas emissions. Out of these emissions up to 28% can be attributed to lighting, heating, and cooling alone. This implies that there is a great potential in optimizing the building performance with novel ways to make buildings carbon neutral. Though major inefficiencies in energy management are caused by the HVAC (heating, ventilation, and air conditioning) systems, artificial lighting also has a significant part in it. In 2020, electricity consumed by lighting is about 6% of total U.S. electricity consumption [3].

The benefits of daylight for building occupants are studied extensively in great detail in several studies [4]. They range from improved health and well-being to increased productivity [5]. However, designing daylight requires careful consideration of its integration into building control systems as it varies in direction, colour, and intensity over time. These variations are the design parameters that are difficult to cope with as they have a considerable impact on both the visual and thermal environment.

A successful daylight design needs to accommodate the trade-offs and optimization between competing aspects such as choice of lighting, facade layout, and space configuration. This necessitates identifying the appropriate optical properties of the glazing system

that provide adequate daylight while avoiding glare and undue heat gains. Shading systems such as electrochromic (EC) glazing have the ability to transition between different light transmission states with the application of little voltage. The transitions in EC glazing are seamless and it provides an unobstructed view during this process. EC glazing can also be divided into several zones and light transmission of each zone can be controlled separately, which gives it an inherent advantage over other shading systems as it can allow a high amount of useful daylight into the visual environment. Moreover, it is also known to provide effective sun protection reducing heating and cooling loads [6].

Designing a control strategy requires a deeper understanding of the problem [7]. In the context of EC glass, choosing the right set of variables for the control is important. A simple approach can be using the visual comfort parameters such as workplace illuminance and glare in choosing the right combination of EC glass. In this case, a simple ranking [8] of the combinations by maximizing workplace illuminance and minimizing glare would suffice, but there is a drawback to this approach. A better explanation for this would be possible with an example. Imagine a scenario where there is an option of electric lighting that can be switched on when there is not enough daylight in the workspace and also consider a situation where the EC glass is completely tinted to dark to avoid glare. In this case, the simple ranking strategy tries to increase the workplace illuminance by switching on the light and bring it to the comfortable range. In the meanwhile, if the sun is covered by clouds, the chance of glare is reduced. In this situation, would the occupants perceive the change in glare, since the EC glass is already darkened to prevent the glare and electric lighting is enough to keep the occupants in a comfortable range? More precisely would the sensors used for detecting glare and illuminance at the workspace be able to understand the external changes? Therefore this control would be stuck in the never-ending loop as it does not consider the external factors. Here the main drawback for it to be a control strategy is its lack of adaptation to the changing sky conditions. After all the sun position (zenith and azimuth) and radiation (direct and diffuse components) are the main drivers that change the illuminance and glare that are perceived in workspaces.

Therefore a good control strategy should understand the changing sky conditions and take an action to change the EC glass to a comfortable combination. As there are a large number of possible sky conditions, a strategy that can learn and adapt to user comfort can be called a real control strategy [9]. Another advantage of using such control is that it can be transferable to different climates with different sky conditions. Even if the climate is fixed, the pattern of sky conditions change continuously. Therefore, it has to be trained with several changing sky conditions and to be allowed to explore the comfort space of the occupants. Reinforcement learning especially the Deep Q learning or Deep Q Network(DQN) agents can be perfect for developing such a control strategy. DQN can converge faster than their policy gradient variants of reinforcement learning and is also suitable when there is a continuous observation/state space (like the changing sky condition and occupant comfort) but only supports a discrete action space. Since our problem deals with a discrete action space i.e., to find the right combination among the possible EC glass combinations, DQN is a perfect choice for this control strategy.

In Section 2.1, the model of the test room is described and its simulation with the help of Radiance [10] is explained in Section 2.2. The data collection from annual simulations is described in Section 2.3, which contains both glare and daylight values. Section 2.4 gives a brief introduction into the DQN agent and how the observations, states, actions and rewards are designed. The training of the DQN agent is described in Section 2.5.5 along with the libraries and hardware used. Section 3 presents the results in various scenarios for daylight availability and glare. The performance of DQN agent in low light, bright light and perfect daylight scenarios are discussed in Section 3.1. The glare reduction capability of the DQN agent at different workstations is discussed in Section 3.2. Section 4 summarizes the DQN agent's capability and the level of enhancement in visual comfort.

## 2. Methodology

### 2.1. Model Description

The model of the test room represents an office space (see Figure 1). It contains a south-oriented fully glazed window with an area of 14 m² and a window wall ratio of 90%. The window is divided into three different zones namely the top, middle and bottom. The room is 6 m in length, 5 m in width, and 3.3 m in height with a floor area of 30 m².

The layout of the room can seat four occupants in each of the four workspaces. Each of the workspaces is suitable for executing common office tasks and the occupant's line of sight is at a height of 1.2 m above ground. The seating configuration of the occupants is shown in Figure 2. The occupants in workspaces 1&4, 2&3 face each other.

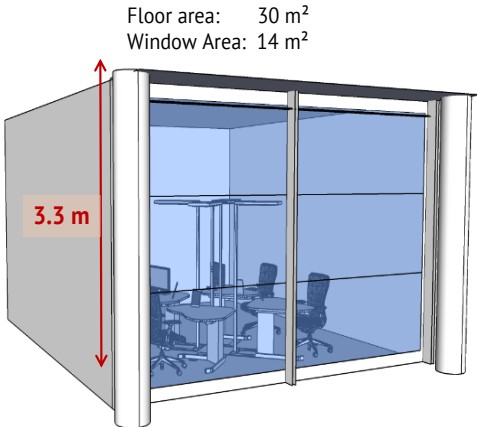

Floor area: 30 m²
Window Area: 14 m²

3.3 m

**Figure 1.** The three different zones of the electrochromic glass fitted to the room.

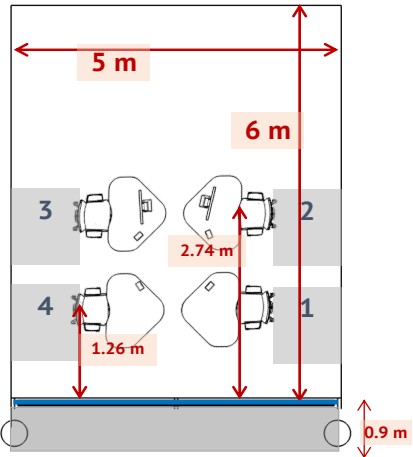

**Figure 2.** The layout of the model office room.

To simulate the exact occupant comfort in this model room, each element that can be seen should be exactly modelled in Radiance (see Section 2.2). This allows to precisely simulate the lighting at each workspace using electrochromic glass. Although a regular test room can be easily modelled and converted into Radiance format using tools like Sketchup with some plugins, it is particularly non-trivial to model an electrochromic glass in radiance.

### 2.2. Simulating Electrochromic Glass in Radiance

Radiance is a set of programs that allows visualization of lighting in design. It has the capability to simulate a glass given its geometry, material, time, date and sky conditions.

Some of the characteristics of glass such as transmittance, front, and backside reflectance, and resulting angular dependencies are key to understanding its performance. Therefore Radiance also uses these properties of the glass to simulate and visualize as close as possible. More information on Radiance simulations can be found in Section 2.3.

A single layer of uncoated basic glass can be modelled optimally in Radiance, as they have the same front and backside reflectances but modelling multiple layered glasses is not straightforward. Multilayered glass is an insulating glazing unit, which is common in most architectural spaces as they have insulating units with various inert gases in the space between layers. Often the surfaces in these composite systems even have performance coatings applied to control the solar gain. While transmittance will be constant in both directions, the front and back reflectances will vary depending on the angle of incident light. Therefore these systems are not only complex but need performance data for each layer of the glazing materials. Usually, this complex modelling and evaluation of glazing systems can be done with Optics, a software developed by LBNL. Optics utilizes the International Glazing Database (IGDB) [11] to obtain the measured spectral and thermal data from a collection of 4200 glazing products.

The glazing system modelled in Optics can be exported in Radiance format but it still needs suitable Radiance material descriptions so that it can be used in simulations. Usually, each layer is exported individually and then assembled into a library with *optics2glazedb*, which is provided by Visarc [12]. The Radiance Glaze script uses this created library and assigns *BRTDfunc* material type to generate the desired glazing unit combination.

### 2.3. Annual Simulations

### 2.3.1. Glare Simulations

The magnitude of glare can only be accurately estimated with user assessments and characterizations together with the physical factors (e.g., solid angle and luminance of the glare source, background luminance, etc.). Several experimental studies were conducted to determine the subjective magnitude of glare discomfort especially with high values such as for intolerable or uncomfortable levels. The daylight glare probability or DGP [13] measures glare caused by daylight. It conveys the fraction of people disturbed by the given daylight situation. It is given by the following equation

$$DGP = 5.87 * 10^{-5} * E_v + 9.18 * 10^{-2} * log(1 + \sum_i \frac{L_{s,i}^2 \, \omega_{s,i}}{E_v^{1.87} \, P_i^2}) + 0.16 \tag{1}$$

where $E_v(lux)$ is the vertical eye illuminance. The other parameters are

- $L_{s,i}$ (cd/m$^2$), the luminance of the glare source $i$. Usually, the luminance of the sky seen through the window is considered here.
- $\omega_{s,i}(sr)$, solid angle of the glare source $i$. Here the apparent size of the visible area of the sky w.r.t to observer's eye through the window is considered.
- $P_i$, Guth's position index of the glare source $i$. It is calculated relative to the position of the observer's line of sight which includes azimuth and elevation of the source.

DGP in contrast to other glare metrics has shown a better correlation (about 0.97) with the observer's responses to the glare perception.

DGP is computed based on the evaluation of a 180-degree fisheye HDR image, which contains the full luminance distribution from all the light sources in the viewing direction. It is implemented in a tool called *Evalglare* which is part of Radiance [10]. DGP generates values between 0 and 1 but the valid values are between 0.2 and 0.8. DGP values less than 0.2 are due to low light and are thus ignored. Table 1 gives the significance of the DGP values concerning user perception of the light source.

**Table 1.** Significance of DGP values.

| Glare Perception | DGP Value |
|---|---|
| Imperceptible | <0.36 |
| Perceptible | 0.36–0.4 |
| Disturbing | 0.4–0.45 |
| Intolerable | >0.45 |

Since it is not practical to collect the glare ratings from the subjects at each of the work spaces for a whole year, simulations [14] can help solve this issue as they can closely represent reality. In this case, we consider the Mannheim climate and its weather file, which represents standardized weather data of this climate. A sky condition can also be defined based on the type of cloud cover but an accurate representation is possible when there is the availability of the diffuse and direct components of radiation values. The weather file usually provides us with this data for every hour else it can be obtained using predictive models from a local or nearby weather station data.

The main part of these simulations involves creating a fisheye HDR from the viewpoint of the occupant in a workspace and evaluating the glare. Such simulations are only possible with Radiance due to its ecosystem of daylighting simulation tools which can create photorealistic images. Some of the important tools in Radiance used in this simulations are *gendaylit*, *oconv*, *rpict*, *evalglare*. Radiance usually needs its own specifically formatted geometric model of the entity, to be simulated. Usually, this model can be obtained by transforming the SketchUp model or similar geometric modelling tool to radiance format using its built-in tools. The workflow in radiance is as follows :

- A *sky* file is created based on the solar direct and diffuse components along with the time, latitude, and longitude. This can be achieved with the tool *gendaylit*.
- The tool *oconv* uses the created sky file and other geometric files of the test room and creates a binary file in *octree* format to be further processed by other radiance tools. An *octree* file is created for each of the viewpoints of the occupant, each time step, and for every combination of EC glass.
- The tool *rpict* is the one that creates the photorealistic simulation of the entity based on given parameters. It can output several different image formats, but the one suitable is an HDR image in fisheye format.
- *Evalglare* uses the *rpict* generated fisheye HDR images to evaluate glare. It outputs DGP value along with several other glare indices.

Though the workflow looks simple and straight forward, there are several challenges in it. At every timestep, simulations for 64 different combinations of EC glass with a particular sky condition are made. This requires generating 64 octrees, each for every combination and sky file. This is repeated for every hour in a year and then fisheye images are rendered for each viewpoint of the occupant. Therefore a total of 8760 × 64 × 4 = 2.2 million fisheye HDR images are generated. These images are then evaluated with *Evalglare* and the obtained DGP values are stored in a database. The simulations are performed using python wrappers on radiance tools with a multiprocessing approach to rendering several HDRs simultaneously. These simulations took about a month on two 64 core XEON E5 2670 servers.

### 2.3.2. Daylight Simulations

Daylight simulations are performed using the Three-Phase Method [15] in Radiance. It is an alternative to the expensive Daylight Coefficient method though it builds on top of it by splitting the flux transfer path into multiple phases. A very good implementation of this method is available in Ladybug–Honeybee [16] with workflows for the annual daylight simulations. The Three-Phase Method supports point in time parametric simulations with complex fenestration systems such as Electrochromic glass. It usually requires generation

of four types of matrices to be generated before the illuminance values are calculated at the workspaces.

- The View (V) matrix contains the flux transfer from the points on the workspace where illuminance is to be calculated to the Electrochromic glass.
- The Transmission (T) matrix contains the flux transfer through the Electrochromic glass.
- The Daylight (D) matrix contains the flux transfer from exterior of Electrochromic glass to the sky.
- The Sky (S) represents the descritized sky vector with 145 patches based on Tregenza sky model(ref)

The resultant matrix is given by

$$E = V * T * D * S$$

where $E$ is an illuminance matrix for the points on the workspaces. The matrices $V$ ($12 \times 145$) and $D$ ($145 \times 145$) are generated with radiance-based workflows involving *rfluxmtx* or *rcontrib*. $V$ is based on the location of the points on the workspace and $D$ are daylight coefficients w.r.t to EC glass and the sky, respectively. The $T$ matrix is a BSDF of the different EC glass states obtained from the Radiance's *genBSDF*. A detailed description of the workflow can be found in a tutorial by Sarith Subramaniam [17]. These simulations also generate a lot of data, as they are run for 8760 h in a year for three points in each of the workspaces, even though this is less expensive than the glare simulations and can be performed overnight on a normal computer.

*2.4. DQN Agent*

Most real-world problems, in general, are complex with high-dimensional state space and possibly continuous. These high-dimensional inputs can be in the form of sensory inputs such as frames, time series, or in our case sky conditions/comfort levels. Neural networks are well suited when dealing with such inputs and can learn an estimate of the model, a policy, or the value function. They can also be incrementally trained to make use of the additional samples obtained while the learning is happening. Therefore the variant of Q-learning that relies on neural networks (deep learning) is called Deep Q learning or Deep Q Network (DQN) [18].

DQN can converge to a global optimum, though not as fast as policy gradient variants, which tend to converge in the local optimum. DQNs are sample efficient (off-policy) and especially with the addition of several features such as double DQN, Prioritized experience replay, duelling networks, distributional and multi-step variants significantly improved the convergence speed and stabilized the performance. When action space is discrete, DQN can be chosen as the starting point to work with as it also supports both continuous and discrete state spaces. Table 2 shows the main differences with other major variants of deep reinforcement learning algorithms.

**Table 2.** Comparision of deep reinforcement algorithms.

| Algorithm | Description | Policy | Action Space | State Space |
|---|---|---|---|---|
| DQN | Deep Q Network | Off-policy | Discrete | Continuous |
| DDPG | Deep Deterministic Policy Gradient | Off-policy | Continuous | Continuous |
| A3C | Asynchronous Advantage Actor-Critic Algorithm | On-policy | Continuous | Continuous |
| TRPO | Trust Region Policy Optimization | On-policy | Continuous | Continuous |
| PPO | Proximal Policy Optimization | On-policy | Continuous | Continuous |

*2.5. Formal Definition*

The Bellman equations [19] can be extended to calculate the optimal Q-value for the given state–action pair as follows:

$$Q^*(s,a) = \mathbb{E}[r + \gamma \max_{a'} Q^*(s',a')] \tag{2}$$

Here $s, a$ are the current state and action, whereas $s', a'$ are the state–action in the next timestep and $r, \gamma$ are reward and discount factor, respectively. The intution behind the Equation (2) is that if the optimal value of $Q^*(s', a')$ is known for all possible actions $a'$, then the optimal strategy is to select the action that maximizes the expected value of $r + \gamma Q^*(s', a')$.

The basic idea here is to update the Q-values iteratively

$$Q_{i+1}(s, a) = \mathbb{E}[r + \gamma \max_{a'} Q_i(s', a')] \tag{3}$$

such equations converge when $Q_i \to Q^*$ as $i \to \infty$. In reality, this approach becomes impractical as the action–value function is estimated seperately for each sequence with out any generalization. Therefore a function approximator is used estimate the action–value function. This can be represented as

$$Q(s, a; \theta) \approx Q^*(s, a) \tag{4}$$

In deep Q learning we use the neural network as the function approximator and $\theta$ represents the weights of the Q-network. This network can be trained by minimizing a sequence of loss functions $L_i(\theta_i)$ that are obtained at each iteration $i$,

$$L_i(\theta_i) = \mathbb{E}[(y_i - Q(s, a; \theta_i))^2] \tag{5}$$

where $y_i = \mathbb{E}[r + \gamma \max_{a'} Q_i(s', a'; \theta_i)]$ is the target for iteration $i$. The parameters of the previous iteration $\theta_{i-1}$ are held fixed when the loss function is being optimized, but the parameters of target $\theta_i$ change as they depend on the network weights. By differentiating the loss function with respective weights $\theta_i$ we arrive at the gradient of the loss function in equation

$$\nabla_{\theta_i} L_i(\theta_i) = \mathbb{E}[(r + \gamma \max_{a'} Q(s', a'; \theta_{i-1}) - Q(s, a; \theta_i)) \nabla_{\theta_i} Q(s, a; \theta_i)]. \tag{6}$$

It is computationally expensive to calculate the full expectations for the above gradient, and therefore this loss function is optimized by stochastic gradient descent. Based on the frequency of the weights being updated the expectations are replaced by the samples from the environment and its action Selection strategies.

The DQN Agent observes the environment then takes an action in it to achieve a reward. The reward achieved depends on the state it reached by taking a particular action in the environment. Here the environment includes both the observations and states, though the DQN agent can only see observations before taking the actions. The states here are merely to calculate the reward for the agent. The following sections explains them in detail.

### 2.5.1. Observations

The observations considered here are solar altitude, solar azimuth, diffuse horizontal irradiance, direct normal irradiance, and occupancy. The solar altitude and azimuth can be calculated based on time, latitude, longitude. Here the Mannheim climate was chosen. The diffuse and direct components are used from the Mannheim weather file. Alternatively, they can also be obtained for any location using predictive models developed by Kalyanam and Hoffmann [20] based on the data from a local weather station. Here diffuse and direct radiation is part of observations because they can represent sky condition pretty much accurately.

The occupancy parameter indicates the presence of occupants in the room. It takes the value 0 (absent) or 1 (present). Usually, only working hours are part of occupant hours. In this experiment 2015 is considered as a reference year and the occupant hours amounted to 2610 h after removing all the holidays and nights.

### 2.5.2. States

The state space for the DQN is both the visual comfort and discomfort of the occupants. The visual comfort/discomfort is defined based on the daylight availability and discomfort glare. The state-space here is continuous based on these two parameters. The comfortable daylight availability in the workplace considered here is between 300 and 3000 lux. Any light level above 3000 lux increases the discomfort as it becomes too bright in a workspace. The imperceptible glare scenario is considered based on the DGP values between 0.2 and 0.36. Any value above 0.36 increases the chance of glare.

The DGP values are obtained from the fisheye images generated from the radiance simulations. Section 2.3.1 explains how the DGP values are generated. Similarly, Section 2.3.2 explains how daylight at a workspace is calculated using radiance simulations.

The DGP values are calculated for each workspace separately, which means the state of each occupant can be different. This results in high dimensional state space. At the same time, the daylight is calculated for 3 points for each workspace. This adds another 12 dimensions to it. Therefore the illuminance value is averaged for workspace 1 and 4 (both near the facade) and workspace 2 and 3 (both away from the facade). This reduces the dimensions of state space to only 6. This can be represented as follows.

$$\langle \langle d_1, d_2, d_3, d_4 \rangle, \langle l_{1,4}, l_{2,3} \rangle \rangle$$

where $d_1, d_2, d_3, d_4$ are DGP values at respective workspaces and $l_{1,4}, l_{2,3}$ are averaged workplace illuminances.

### 2.5.3. Actions

The actions taken by DQN are about changing the transmission of EC glass. As the EC glass considered here has 3 zones and each zone can transmit light in 4 different transmission levels, a total of 64 ($4 \times 4 \times 4$) possible combinations are obtained in which light can be transmitted at different intensities. Each action taken by the DQN agent is perceived by the occupants differently which results in different DGP values and workspace illuminances. The actions i.e., the combinations of the EC glass are hereafter represented as

$$\langle TopZone, MiddleZone, BottomZone \rangle$$

For example, combination 012 representing the top zone is clear, the middle zone is at low tint and the bottom zone is at medium tint. Table 3 shows the corresponding IDs of each tint of EC glass.

**Table 3.** Properties of various states of electrochromic glass and their IDs.

| State | ID | IGDB ID | $T_{vis}$ |
|---|---|---|---|
| Clear | 0 | 8905 | 0.448 |
| Low tint | 1 | 8906 | 0.121 |
| Medium tint | 2 | 8908 | 0.040 |
| Dark | 3 | 8909 | 0.007 |

### 2.5.4. Reward Function

The DQN agent achieves a reward or penalty based on how good or bad the state it has reached is. The ability of the DQN agent to converge lies in choosing a proper reward function. The reward function that can reward or penalize gradually based on some criteria helps achieve DQN agent converge quickly and choose meaningful actions.

The criteria of the reward function are chosen based on the state of the occupant's visual comfort. The reward gradually increases from the DGP value of 0.2 until 0.36 then starts penalizing once it crosses 0.36. Likewise, it also penalizes when the DGP value goes under 0.2. The same happens with daylight availability in the range of 300 to 3000 lux. Both daylight availability and Glare are equally important so they are given equal weightage in reward and penalty.

The mathematical representation of the reward function for glare is given by:

$$r_g(d_i) = \begin{cases} (\frac{d_i}{d_{min}}) - 1, & \text{if } d_i \leq d_{max} \\ -\frac{d_i}{d_{min}}, & \text{otherwise} \end{cases}$$

where $d_i$ is the DGP value at each of the workspaces from 1 to 4 and $d_{min}$ is the minimal value of DGP where the comfort range starts i.e., 0.2, similarly $d_{max}$ is the maximum value of DGP where teh comfort range ends i.e., 0.36. The total reward based on DGP is

$$R_{glare} = \sum_{i=1}^{4} r_g(d_i)$$

The reward function for daylight availability is given by:

$$r_{ill}(l_i) = \begin{cases} (\frac{l_i - l_{min}}{l_{max} - l_{min}}), & \text{if } l_i \leq l_{max} \\ -\frac{l_i}{l_{max} - l_{min}}, & \text{otherwise} \end{cases}$$

Figure 3 shows the graphical representation of the glare reward function. The total reward from glare can vary from $-5$ to 3.2.

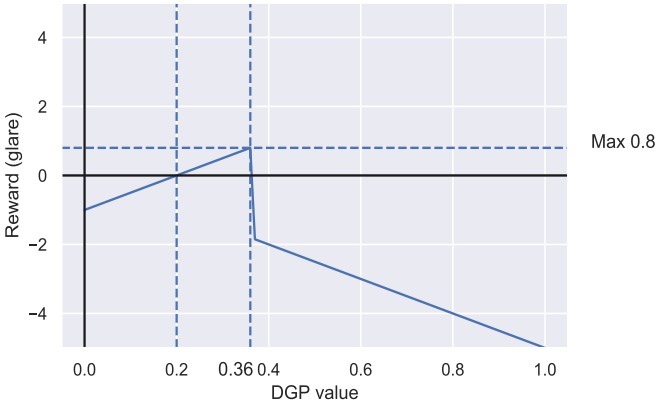

**Figure 3.** Reward function based on glare (DGP values).

The total reward based on useful daylight availability is

$$R_{ill} = \sum_{i} r_{ill}(l_i)$$

where $l_i$ is the average illuminance value at each of the workspaces near (1 and 4) and away (2 and 3) from the facade. $l_{min}$ is the minimal illuminance value of daylight where the comfort range starts i.e., 300 lux, similarly $l_{max}$ is the maximum illuminance value of daylight where comfort range ends i.e., 3000 lux.

Figure 4 shows the graphical representation of the illuminance reward function. The total reward from illuminance can vary from $-\infty$ to 4.

Therefore the total reward achieved by the DQN agent at any time step is given by

$$R = R_{glare} + R_{ill}$$

Theoretically, the total reward can vary from $-\infty$ to 7.2.

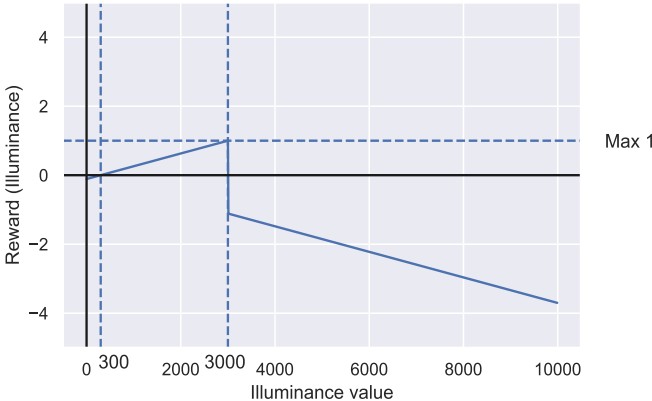

**Figure 4.** Reward function based on illuminance (Workplace illuminance).

### 2.5.5. Training the DQN Agent

The DQN agent's major task is to select optimal actions to maximize the rewards. It observes the environment i.e., sky condition and takes an action (one of 64 EC glass combinations) based on the estimates of observation–action pair or the Q-value. The action selected by the DQN agent is rewarded or penalized based on how good or bad the selected combination of EC glass is based on the comfort of the occupants(state-space). Figure 5 shows the overview of the DQN agent's observation space, state space, actions space, and reward space. Since the rewards, in this case, are not a discrete number but a continuous range based on each parameter of the state space is called a reward space.

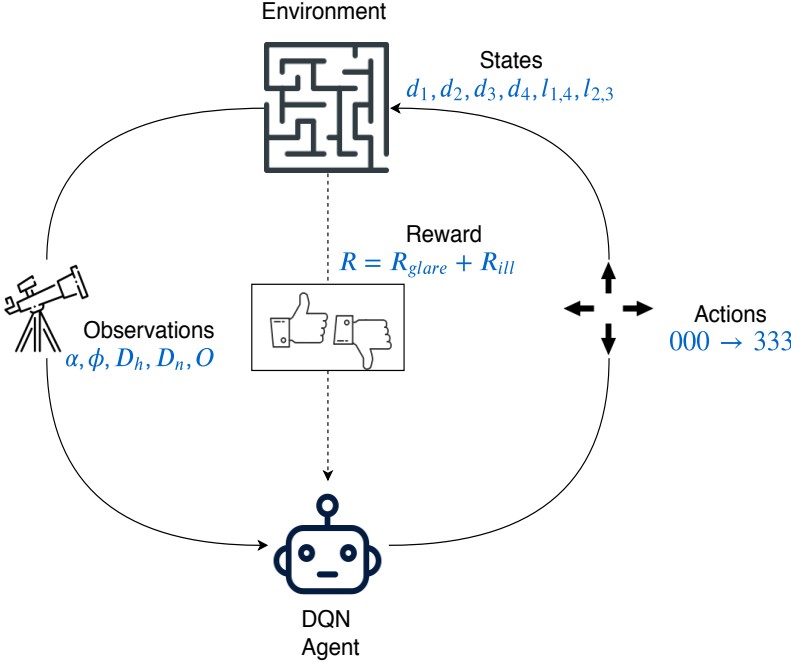

**Figure 5.** DQN agent with observations, states, rewards and actions.

Since the DQN needs to estimate Q-values which basically depend on previous Q-values and so on, DQN starts with a random set of Q-values in the beginning. Therefore initially most of the actions chosen are random which results in either better or worse Q-value for the future. Though the observation space is small considering only 8760 h in a year the state space is huge and high dimensional with 4 DGP values and 2 illuminance values for each of 64 combinations for an entire year. This increases the number of Q-values astronomically as they are simply a kind of database with a mapping of all sky conditions

and EC glass combinations. In this case, the number of Q-values can go as high as up to 2.2 million (8760 observations × 64 combinations × 4 viewpoints). Now the problem can be represented as predicting the Q-values for each EC glass combination separately given the set of sky conditions. Neural networks are famous for solving such complex problems very easily. Therefore a neural network architecture that takes the sky condition as inputs and spits out the Q-values for each EC glass combination is chosen. The EC glass combination with the maximum Q-value is chosen for the reward calculation.

Employing a neural network simplifies the task at hand but it brings along several parameters that are to be tuned to optimize the network. These are hyperparameters that include the number of layers in the neural network, number of neurons in each layer, batch size, and number of steps to be trained. Apart from the hyperparameters of the neural network, there are few parameters specific to the DQN agent that are to be tuned such as the learning rate, discount factor, and rate of exploration. Each of the hyperparameters is searched either manually with a given set of values or randomly in a range of numbers. Therefore a grid search over these hyperparameters yields several configurations in which the DQN agent can be trained. In our case, with the given number of combinations, there are over 1200 possible configurations possible that are being trained to find the best DQN Agent i.e., highest reward achieving agent. Therefore, a good tuning algorithm is needed that can schedule a training course for each DQN agent as a job that can be monitored. Apart from these, an EC glass environment is set up inside OpenAI gym [21], which provides a framework to develop environments and test the models.

RLlib [22] is a good library that contains highly efficient implementations of reinforcement learning algorithms and supports high performance and scalability. It is part of the Ray project [23] which also supports Tune [24], which can be used for scalable hyperparameter tuning. Both RLlib and Tune are built on top of the Ray engine, which offers high scalability with unified API for experimentation with a variety of applications. Tune offers implementations of several state-of-the-art hyperparameter search algorithms. Asynchronous Hyperband is one such algorithms that implements an asynchronous version of successive halving originally implemented in the Hyperband scheduler. The below piece of code shows the usage of this scheduler.

Here the scheduler aggressively early stops the underperforming configurations and allocates more resources to the promising configurations. It reduces the number of configurations by half for every rung, i.e., every 100 training steps. Figure 6 shows both the early stopped configurations and successfully continuing configurations based on the criteria with minimal mean temporal difference error (mean TD error). Over a period it allows only the best-performing configurations to train. This makes the training complete faster even considering a huge number of configurations. Figure 6 shows a minimalistic representation of configurations to show the ones with high mean TD error are terminated. Here the replay buffer size considered is about 1 million samples of Q-values. This value is selected after experimenting with several sizes.

The DQN models are checkpointed at every step and saved. The actual training is performed on DGX-2 in a High-performance cluster (Elwetrisch) at the University of Kaiserslautern. It took about a week to complete the training but the best configurations are filtered out in the first 72 h. All the configurations are allowed to train for 876,000 steps with an annealing exploration rate of over 700,000 and reaching a final exploration rate of $5 * 10^{-3}$. In the remaining steps, the models allowed rolling out with a constant exploration rate to capture any good checkpoints that may come across.

The best configurations are chosen based on the highest reward achieved with both glare and illuminance criteria. Figures 7 and 8 show the rewards gradually achieved over the course of the training and the Table 4 shows the absolute values.

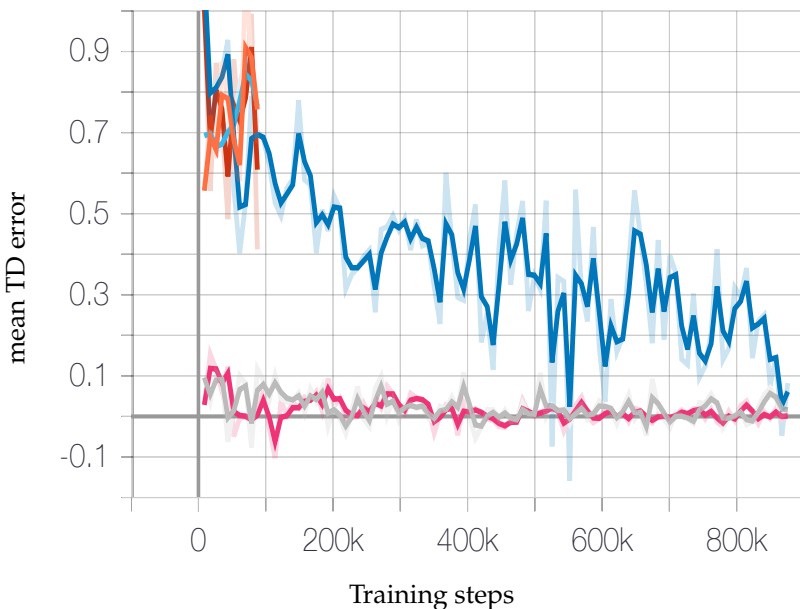

**Figure 6.** Asynchyperband successive halving approach to stop underperforming jobs early (random configurations).

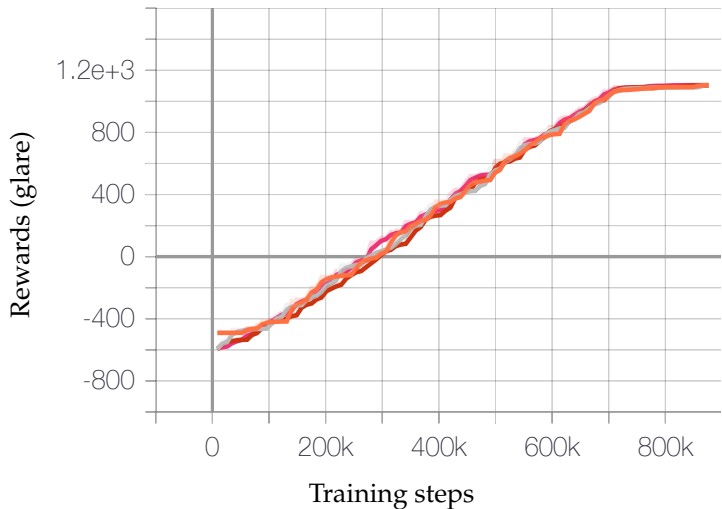

**Figure 7.** Top jobs achieving rewards (glare) over the training duration.

**Table 4.** Best configurations and their corresponding maximum rewards.

| | ID | Reward (Glare) | Reward (Illuminance) | Total Reward | Network Config |
|---|---|---|---|---|---|
| ● | A | 1102 | 626 | 1728 | [2048, 2048] |
| ● | B | 1106 | 625.6 | 1731.6 | [2048, 2048] |
| ● | C | 1108 | 631 | 1739 | [2048, 1024] |
| ● | D | 1104 | 628.5 | 1732.5 | [4096, 2048] |

These four configurations almost arrive at the maximum reward with slight differences but the configuration with minimal neurons is chosen (configuration 'C'). The problem is considered solved when the number of glare hours is less than 5% of the occupied hours in any of the workspaces. The best performing DQN agent had about 3% glare hours.

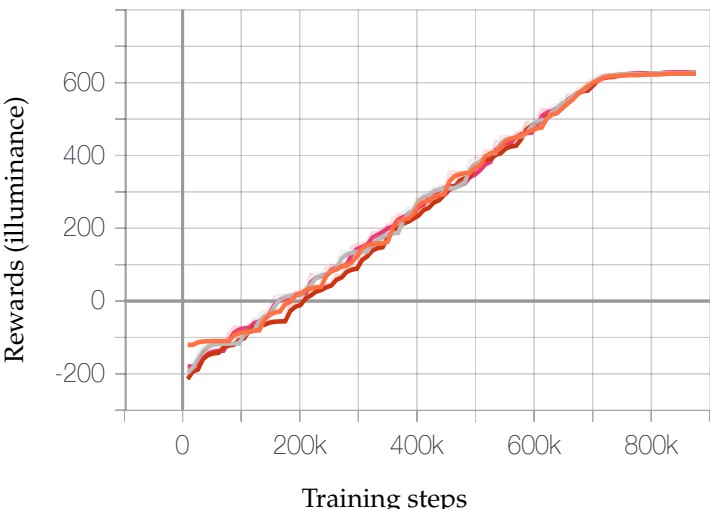

**Figure 8.** Top jobs achieving rewards (illuminance) over the training duration.

## 3. Results & Discussion

As discussed in Section 1 there are only benchmarks but not a fullfledged control for EC glass at this moment. To compare the results, we use the most frequent combinations achieved by the DQN agent to compare, if they were chosen instead of the agent's preferred combination. The comparison is based on the annual data of each combination if they were chosen constantly.

It can be observed from Figure 9 that 000 is the most frequently chosen combination by the DQN agent. The next frequent combinations are 010 and 001 which are only of 1/3 frequency that of 000. One thing which is common in all these combinations is the clear case '0' which is predominant. The reward function we defined pushed the DQN Agent to choose those combinations that allow more light. In the next sections, we dive deeper into the results achieved by the DQN agent to verify if our two main important goals are achieved, i.e.,

- Maximizing the daylight availability;
- Minimizing the glare.

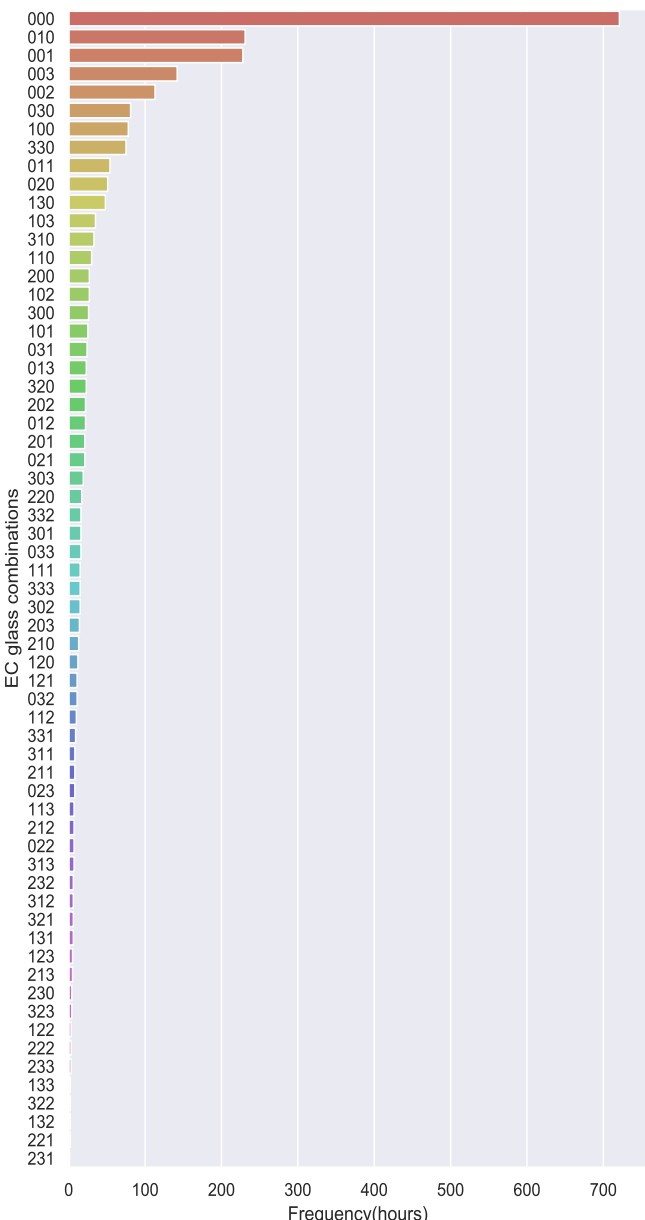

**Figure 9.** Frequency of EC glass combinations chosen by DQN.

*3.1. Daylight at Workspaces*

3.1.1. Lowlight Scenario

One of the goals of this DQN algorithm is to let in enough daylight by EC glass to all the workspaces. Figure 10 shows the low light hours for the two extreme combinations of EC glass along with the selected choices by the DQN agent. It can be observed that 000 has the lowest and 333 has the highest possible low light hours. This implies the combination of EC glass that can provide maximum daylight is 000, which is all clear. Even if EC Glass is left in 000 states for a whole year, it doesn't provide enough useful daylight. It can be seen in Figure 10 that the combination of 000 has low light hours(<300 lux) of about 15% in Workspace 1 and 4, which are close to facade and 22% in Workspace 2 and 3, which are at a distance of 3 m from the facade. These amounts of hours cannot be reduced unless artificial lighting is added to the workspaces.

Using the EC glass in an always clear (000) combination for the entire year defeats the sole purpose of EC glass altogether and, furthermore, this combination can cause a lot of glare which is explained in Section 3.2. Any combination other than 000 (see Figure 11),

with one of the zones tinted to a slightly higher level than clear, can increase the low light hours enormously. The combinations 001, 010, 100 have low light hours of about 59%, 69%, 59%, respectively, which means they make the daylight situation even worse at the workspaces. Any sequence of combinations that utilizes mostly 000 and any darker tints occasionally to reduce glare can provide better daylight. The DQN algorithm provides exactly the right set of combinations to reduce low light hours. The low light hours by the DQN are only a little higher than the 000 combination, i.e., 4% higher at workspaces 1 and 4 and 7% higher workspaces 2 and 3, respectively.

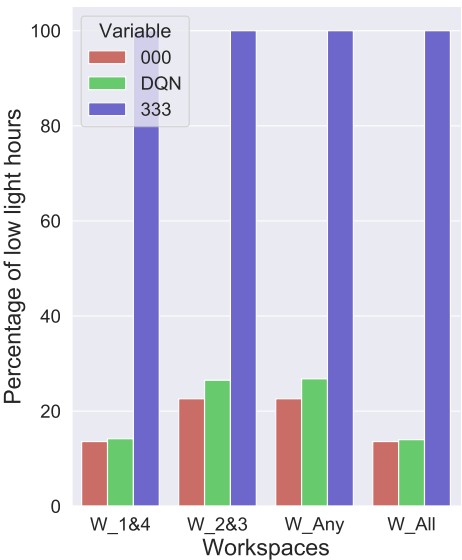

**Figure 10.** Low light hours at all the workspaces with extreme combinations of EC glass.

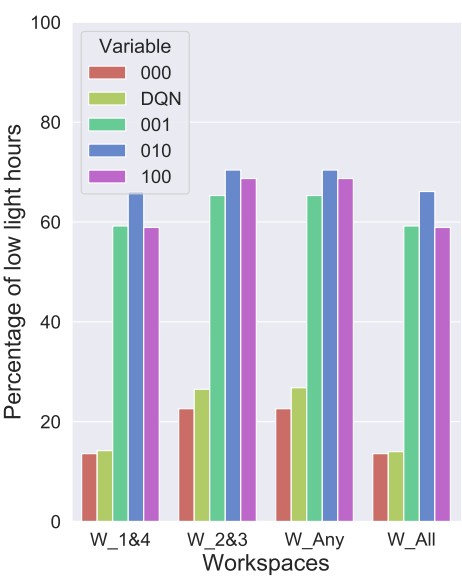

**Figure 11.** Low light hours at all the workspaces with clearest combinations of EC glass.

### 3.1.2. Brightlight Scenario

As in the low light scenario, we compare the extreme cases first, to see bright light hours. The combination 000 will have the highest and 333 will have the lowest number of bright light hours. The combination 333 lets almost no light inside as it transmits around only 1% of the incident light. Therefore, it makes sense to compare the sequence of combinations from the DQN agent with those combinations that have at least one clear zone. Here we consider combinations 330, 303, 033 for reference. It can be observed in

Figure 12 that keeping the EC glass in 000 can cause bright light for 34% of the time. The combinations 303 and 033 lets in more bright light when compared to 330, where it is minimal. It can be seen from the figure that the set of combinations from DQN agent also has bright light as minimal as 330, i.e., about less than 0.5%.

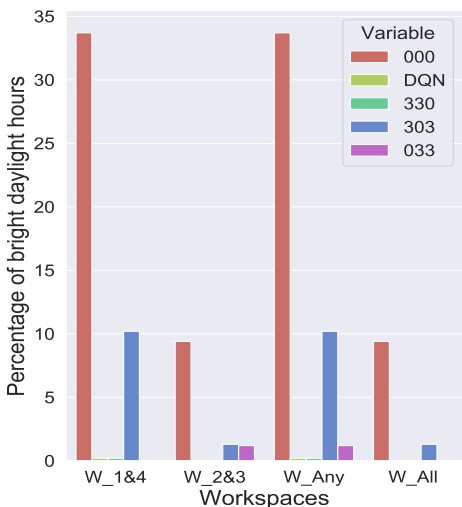

**Figure 12.** Comparison of bright daylight hours obtained by DQN with extreme cases. Observe the extremely low bright daylight hours for the DQN agent at each of the workspaces.

### 3.1.3. Perfect Daylight Scenario

Occupants have utmost comfort when they have the workplace illuminances between 300 and 3000 Lux. These are perfect daylight scenarios with high availability of the daylight that can be properly used. To know if the DQN agents combination achieved maximum daylight availability, let us compare this with the top four frequent combinations chosen by the Agent.

Figure 13 shows the combinations of 000, 001, 010, 003. Of these combinations, 000 makes available maximum useful daylight of only 43% in workspaces 1 and 4 and 78% in workspaces 2 and 3. The combinations 001, 010, 003 can only provide useful daylight in 22–25%, 33–38% in workspaces 1 and 4 and workspaces 2 and 3, respectively. It can be observed that changing any of the zones of the glass in combination 000 reduces the available useful daylight almost by half. In contrast, the DQN agent effectively uses a set of combinations EC glass to provide useful daylight about 75% in workspaces 1 and 4 and 90% in workspaces 2 and 3.

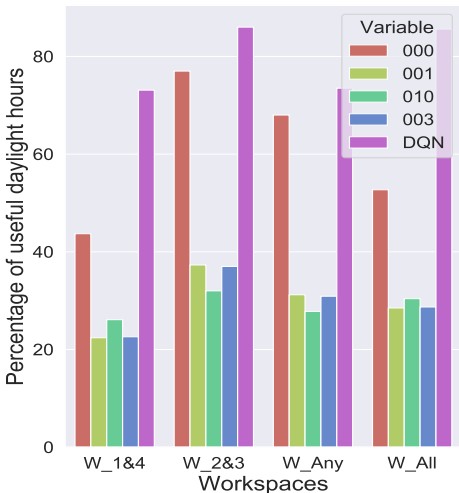

**Figure 13.** Comparison of perfect daylight hours from the top 4 most frequented combinations by rhe DQN Agent against DQN combinations.

### 3.2. Glare Reduction

Allowing useful daylight into space not only makes it comfortable for the occupants but sometimes also uncomfortable due to glare. Figure 14 shows the percentage of glare hours observed by combinations 000 and 333. These combinations are chosen because these are the extremes where the glare is either the highest or lowest due to their clearest and darkest states. It can be observed that there is a 40% occurrence of glare for any of them if only 000 combinations are used. On the other hand, even the combination 333 has about 3% occurrence of glare which means even the darkest combination cannot completely reduce the glare. So any set of combinations that can reduce glare as good as combination 333 would be most suitable for control. The choice of combinations from DQN agent exactly offers this kind of control and sometimes even better.

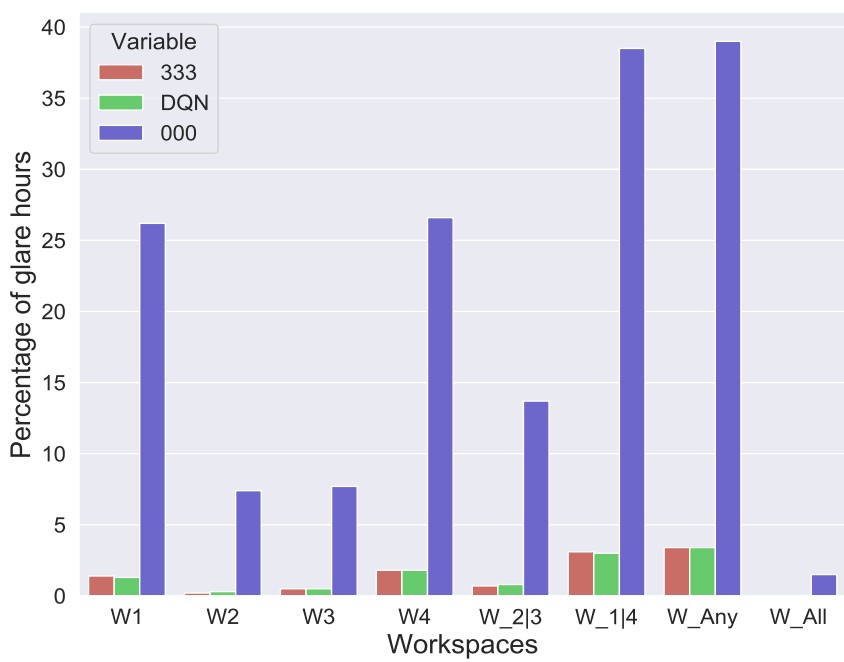

**Figure 14.** Comparison of glare hours with extreme cases of EC glass with the approach of the DQN agent.

It can be observed in Figure 14 that the occurrence of glare hours by the DQN Agent's combinations and 333 are almost the same. In some cases, combination 333 has more glare hours than the DQN agents' combinations in workspace 1. This is a particular scenario that is not expected to happen, as combination 333 restricts more light than any other possible combination. A deeper understanding of these scenarios is required to know how the DQN agent's chosen combinations performed better than combination 333 in avoiding glare.

Figure 15 shows the comparison of combination 333 with the DQN agent-suggested combinations such as 032, 120, 130, 131, 320, 321, 330 for different instances and their reported DGP values in those instants. It can be observed that 333 reported higher DGP values than other combinations, though it should be the other way around. This is an unusual phenomenon but can be explained due to the role of contrast in the glare. Each of the DQN Agent-suggested combinations have a clear or slightly clear zone that allows some light to pass through and illuminate around the glare source uniformly, where combination 333 restricts almost all the light. This scenario creates a high contrast between the glare source and the surrounding, which makes even a small intensity glare source more pronounced. A similar analogy can be observed with glare in the night driving. The headlights are not a glare source during the daytime but only during the night time due to their high contrast in the darkness. They can also be mathematically explained from the DGP equation in (sec ref). The vertical illuminance ($E_v$) in the denominator of the second part of the equation reduces drastically when combination 333 is chosen. This effect becomes much more intense when $E_v$ is less than 1 lux and then raised to the power of $a_1$, which equals greater than 1. Having a log to this term reduces the effect considerably yet it pushes for a value above the glare threshold of 0.36.

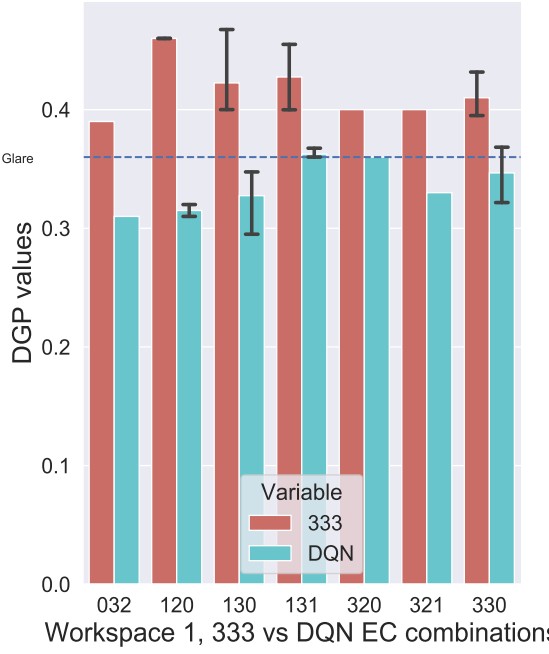

**Figure 15.** Workspace 1 : Comparison of 333 (dark) combination with DQN-proposed combinations.

The aforementioned situation is more pronounced in Workspace 4 which can be observed in Figure 16, as the probability of direct sunlight in the visual field is more for the occupants in this workspace. As a result, several new combinations are chosen by the DQN Agent such as 010, 022, 111, 201, 300, 302, 310, 313, and 331. These new combinations have one of the zones set to clear (0) or slightly tinted (1) to decrease the contrast, thus reducing the chance of glare.

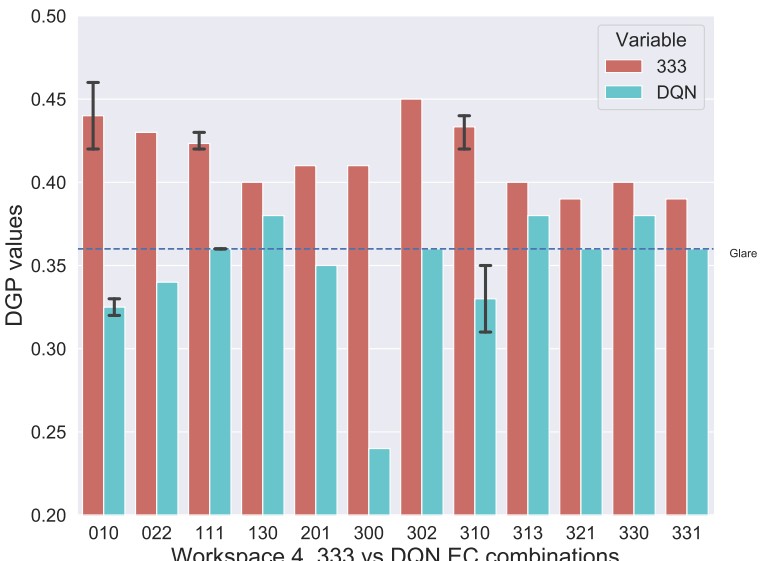

**Figure 16.** Workspace 4 : Comparison of 333 (dark) combination with DQN proposed combinations.

### 3.3. Analysis of DQN Agent's Annual Results

3.3.1. Annual Glare with DQN Agent

Though the DQN agent performs better than the darkest combination 333 in reducing the glare, still there are certain scenarios where it cannot completely avoid it. Figure 17 shows the DQN agent's performance over the year from January to December (0 to 8760 h). It can be observed (DGP values above 0.36) that for all the positions the glare occurs mostly at the beginning and end of the summer, i.e., during spring and fall seasons. This can be attributed to the fact that Solar altitudes are lower in these periods. Workspaces 1 and 4 have more glare compared to 2 and 3 due to they being close proximity of EC glass and having more view towards the sky.

Workspace 1 has less glare when compared to workspace 4 though is at the same distance from EC glass, as the sun arrives first in the latter's viewpoint. When the sun moves into the field of view of the observer in workspace 1, the sun is at a higher altitude to cause any significant glare to the occupants in this workspace. There is also a high chance of glare in summer due to the bright clouds but the DQN Agent chooses good combinations of EC glass and manages the DGP value in the comfortable range.

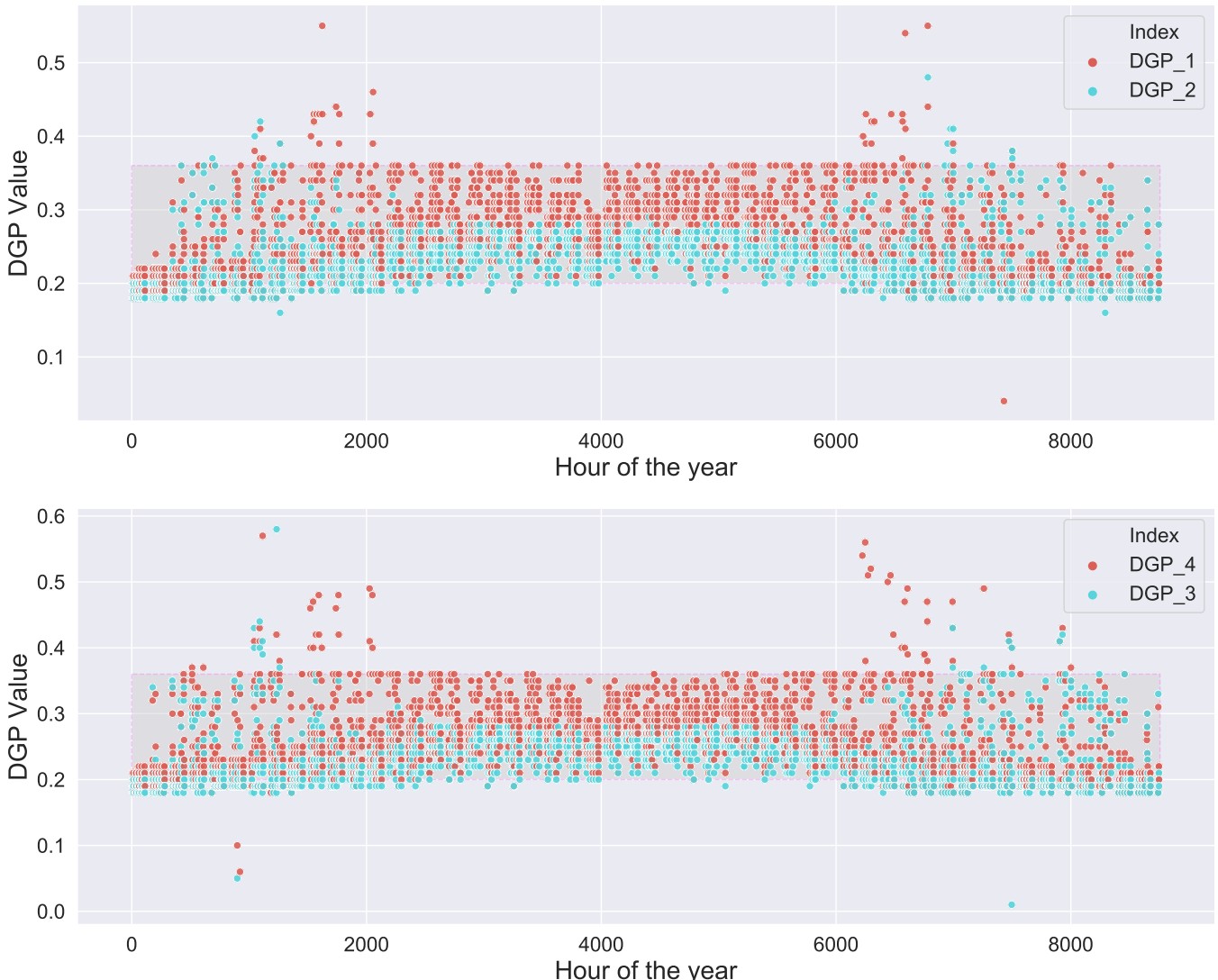

**Figure 17.** Glare occurrence for all the during the course of the year with the DQN agent's control.

3.3.2. Annual Daylight Availability with DQN Agent

Although the DQN Agent can select the best actions to maximize the useful daylight, there are several situations where there is low light, especially from mid fall to mid spring. Figure 18 shows the illuminance values over the whole year. It can be observed that in workspaces 2 and 3 which are a bit further from the facade received comparably less daylight during the colder months (see the orange values less than 300 lux), though workspaces 1 and 4 almost always have better daylight.

There are also some situations where workspaces 2 and 3 receiving more daylight than workspaces 1 and 4 in non-summer months, which can be attributed to the DQN agent's control to prevent glare at the cost of some reduced daylight but remaining in the acceptable range.

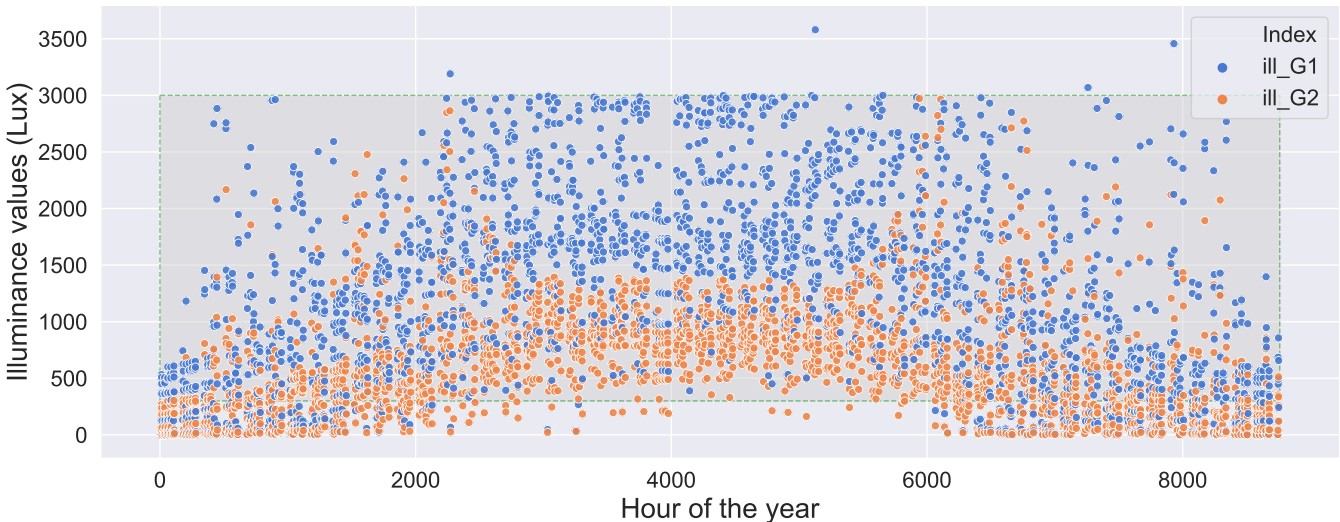

**Figure 18.** Illuminance values obtained with the DQN agent suggested combinations over a year.

### 3.3.3. Rewards

The primary objective of the DQN agent is to maximize the rewards, and therefore it tries to be greedy at every timestep but the environment punishes the agent whenever the action is bad. The best action, i.e., the best combination chosen by the DQN agent sometimes doesn't help it to achieve a positive reward, although it receives the maximum reward. This can quite happen when all the combinations are relatively bad. One such case can be the low light conditions, which happen predominantly during winters.

Figure 19 shows the combined rewards achieved by the DQN agent for both daylight maximization and glare reduction. It can be observed that there are several negative rewards during the start (0–2000 h) and end of the year (above 6000 h). Although the DQN agent could reduce the glare and bright daylight hours significantly, the low light hours caused a significant penalty.

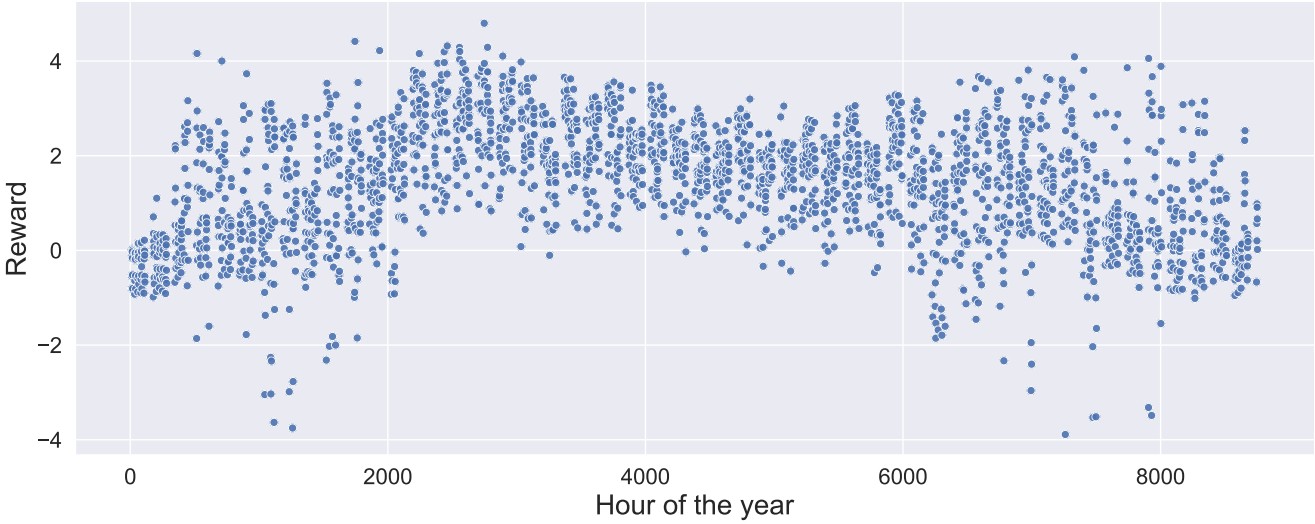

**Figure 19.** Rewards achieved by the DQN agent at different time steps over the course of year.

## 4. Conclusions

Natural light is important for well-being. It not only improves productivity but reduces energy bills. So the objective of this study is to maximize the usage of available daylight. Though natural light has many advantages, inefficient control of it can lead to

visual discomfort to the occupants. So intelligent control natural light coming through fenestration system is important. Although the primary reason to install a shading system is to prevent glare. Apart from glare it also helps in preventing overheating due to high-intensity solar radiation. It not only reduces lighting load but can also reduce the high cooling load in summers and heating load in winters, thus saving energy costs. It improves overall comfort by allowing us to obtain the benefits of natural light at the same time.

A shading system such as Electrochromic glass has some extra advantages like being able to control the intensity of its opaqueness from being almost dark to completely transparent and also the individual zones thus giving more flexibility in allowing the sunlight into workspaces. Electrochromic glass also appears aesthetically pleasing when compared to regular blinds thus improving the mood.

In our study, the Electrochromic glass can be switched to four states with varying transmittance. In the clearest state, it transmits about 68% of the incident visible light, whereas in the darkest state it transmits only 1% of the visible light. Since the Electrochromic glass can be divided into zones and each zone can be switched to different levels of transmittance, we can adjust these zones in such a way that we obtain only the required amount of daylight. The goal of this research is to choose the right combination w.r.t the current sky condition such that glare and bright light can be avoided. In addition, keeping in context here is the viewpoint of the observer, as when there is more than one observer in the room, i.e., several viewpoints, then the feedback from all the observers are to be considered for effective control.

Since the automation of the electrochromic glass needs a feedback loop to learn, reinforcement learning suits very well in this context. This is because we can formulate a specific reward function that suits the problem. In our problem, the reinforcement learning agent first observes the environment with the help of sun position, solar radiation, and workplace occupancy and tries to take any action, by switching to one of the 64 combinations of Electrochromic glass. As a result, the state of the environment changes, either improving visual comfort or making it worse for the occupant. The control parameters for visual discomfort are glare and bright daylight, which can be quantified with metrics like DGP and vertical eye illuminance.

A simulated environment of fisheye HDR images is set up with the help of Radiance and evaluated using Evalglare. The evaluated values are used to reward or penalize the agent. A complex reward function that suits this problem is derived based on several trials and the agent's behaviour. Training the RL models is performed with the data from the weather file for the German city of Mannheim. A neural network is chosen as the learning model to accommodate the huge number of observations, states, and action combinations. Since training involves conducting several experiments in parallel and also optimal usage of resources asynchyperband tuning algorithm is used to tune the experiments.

The results are evaluated separately based on the availability of useful daylight and glare reduction. In both cases, the DQN agent performed better than the benchmarks taken.

- In the case of bright daylight hours, it performs as well as its benchmark combination 330, whereas in the useful daylight scenario it does better than its benchmark combination 000 allowing up to 90% daylight.
- In the case of glare reduction, it performs as well as the darkest combination but allowing more light at the same time.
- The strategy of the DQN agent, i.e., changing different combinations continuously is efficient and reduced glare about 97% of the time and increased daylight availability in 90% of the work hours.

The DQN agent also found some instances in which it chose even better combinations to prevent glare than the darkest state. It understood the role of contrast in glare automatically and adapted its policy to reduce the contrast in the room, to reduce glare.

**Author Contributions:** Conceptualization, R.K. and S.H.; methodology, R.K.; software, R.K.; validation, R.K. and S.H.; formal analysis, R.K.; investigation, R.K.; resources, S.H.; data curation, R.K.;

writing—original draft preparation, R.K.; writing—review and editing, R.K. and S.H.; visualization, R.K.; supervision, S.H.; project administration, S.H.; funding acquisition, S.H. All authors have read and agreed to the published version of the manuscript.

**Funding:** This work was partially supported by the LiSA project funded by BMWi (EnOB) under Grant No. 03ET1416A-F.

**Conflicts of Interest:** The authors declare no conflict of interest. The funders had no role in the design of the study; in the collection, analyses, or interpretation of data; in the writing of the manuscript, or in the decision to publish the results.

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
