# Peer review of "A Reinforcement Learning-Based Approach to Automate the Electrochromic Glass and to Enhance the Visual Comfort"

_applsci, doi:10.3390/app11156949_

Round 1

Reviewer 1 Report

This issue is interesting and worthy of investigation. The article has a very good structure. Before the possibility to publish the paper, I suggest to realise following modifications:

  • in the introduction part of the article please provide more relevant research articles focused on the issue of lighting, e.g.: Saraiji, R., Al Safadi, M. Y., Al Ghaithi, N., & Mistrick, R. G. (2015). A comparison of scale-model photometry and computer simulation in day-lit spaces using a normalized daylight performance index. Energy and Buildings, 89, 76-86.; Dupláková, D., Flimel, M., Duplák, J., Hatala, M., Radchenko, S., & Botko, F. (2019). Ergonomic rationalization of lighting in the working environment. Part I.: Proposal of rationalization algorithm for lighting redesign. International Journal of Industrial Ergonomics, 71, 92-102.; Dupláková, D., Hatala, M., Duplák, J., Knapčíková, L., & Radchenko, S. (2019). Illumination simulation of the working environment during the testing of cutting materials durability. Ain Shams Engineering Journal, 10(1), 161-169 etc.
  • provide modification dimensioning in Figure 1 according to the standards
  • please provide more detailed information about the results presented in Figure 16-19
  • please provide obtained results in the Conclusion part of the article in bullets

Author Response

Dear reviewer,

Thank you very much for taking time out to review my article. I am very much pleased with your comments and also made the required modifications accordingly. Please find below a point to point clarification about the changes.

  • in the introduction part of the article please provide more relevant research articles focused on the issue of lighting, e.g.: Saraiji, R., Al Safadi, M. Y., Al Ghaithi, N., & Mistrick, R. G. (2015). A comparison of scale-model photometry and computer simulation in day-lit spaces using a normalized daylight performance index. Energy and Buildings, 89, 76-86.; Dupláková, D., Flimel, M., Duplák, J., Hatala, M., Radchenko, S., & Botko, F. (2019). Ergonomic rationalization of lighting in the working environment. Part I.: Proposal of rationalization algorithm for lighting redesign. International Journal of Industrial Ergonomics, 71, 92-102.; Dupláková, D., Hatala, M., Duplák, J., Knapčíková, L., & Radchenko, S. (2019). Illumination simulation of the working environment during the testing of cutting materials durability. Ain Shams Engineering Journal, 10(1), 161-169 etc.

Response

Thanks for pointing out the respective articles. They are included in the lines 32,48 and 161 accordingly.

  • provide modification dimensioning in Figure 1 according to the standards

Response

The Figure is modified accordingly

  • please provide more detailed information about the results presented in Figure 16-19

Response

A more detailed explanation is added in the lines 494-499, 502-515, 517-526, 528-536

  • please provide obtained results in the Conclusion part of the article in bullets

Response

Presented the results in in bullet points in lines 586-593

Reviewer 2 Report

The provided study on electrochromic glass automation control is very interesting and promising. The paper is well written and very detailed.

Some general comments which should be addressed in the paper:

Units of the parameters should be added in the lines 145-151

Figure 11 seems to be missing some data? Or is it so little that it is not visible in the figure?

How can the theoretical research be implemented in real word scenarios? What happens if the workplace locations are changed during the usage of the office? Does the 2 month long simulation process needs to be redone?

Author Response

Dear reviewer,

Thank you very much for taking time out to review my article. I am very much pleased with your comments and also made the required modifications accordingly. Please find below a point to point clarification about the changes.

  • Units of the parameters should be added in the lines 145-151

Response

Added the units for better clarity. Please note that Guth’s index doesn’t have any units as it is position index.

  • Figure 11 seems to be missing some data? Or is it so little that it is not visible in the figure?

Response

Your assumption is right, it is so little that it is not visible in the figure. We have to zoom in a bit to see the bright light hours caused by DQN agent with its control. It prevented the bright light hours most cases successfully.

  • How can the theoretical research be implemented in real word scenarios? What happens if the workplace locations are changed during the usage of the office? Does the 2 month long simulation process needs to be redone?

Response

Thanks for asking a very practical question. In the current configuration the illuminance is measured at the eye level with a fixed position but in practical setting the feedback sensor can be positioned behind the occupant for example on top of the chair’s head rest to get continuously accurate measurements based on the occupant’s direction of view. Since the model learns to adapt based on the feedback from the sensors/occupants, it eventually learns the new preferences, but the model has to be retrained with new data occasionally (for example automatic nightly/weekly training in the cloud with new data).

Round 2

Reviewer 1 Report

Dear authors!

Thank you very much for your revision. In my opinion, this article is possible to publish in current form.